# CryoSat-2 Significant Wave Height in Polar Oceans Derived Using a Semi-Analytical Model of Synthetic Aperture Radar 2011–2019

Harold Heorton [1,*,†], Michel Tsamados [1,†], Thomas Armitage [2,†], Andy Ridout [1,†] and Jack Landy [3,4,†]

1 Centre for Polar Observation and Modelling, University College London, London WC1E 6BS, UK; m.tsamados@ucl.ac.uk (M.T.); a.ridout@ucl.ac.uk (A.R.)
2 Jet Propulsion Laboratory, California Institute of Technology, Pasadena, CA 91125, USA; tom.w.armitage@gmail.com
3 Bristol Glaciology Centre, School of Geographical Sciences, University of Bristol, Bristol BS8 1TH, UK; jack.c.landy@uit.no
4 Earth Observation Group, Department of Physics and Technology, UiT The Arctic University of Norway, 9037 Tromsø, Norway
* Correspondence: h.heorton@ucl.ac.uk
† These authors contributed equally to this work.

**Abstract:** This paper documents the retrieval of significant ocean surface wave heights in the Arctic Ocean from CryoSat-2 data. We use a semi-analytical model for an idealised synthetic aperture satellite radar or pulse-limited radar altimeter echo power. We develop a processing methodology that specifically considers both the Synthetic Aperture and Pulse Limited modes of the radar that change close to the sea ice edge within the Arctic Ocean. All CryoSat-2 echoes to date were matched by our idealised echo revealing wave heights over the period 2011–2019. Our retrieved data were contrasted to existing processing of CryoSat-2 data and wave model data, showing the improved fidelity and accuracy of the semi-analytical echo power model and the newly developed processing methods. We contrasted our data to in situ wave buoy measurements, showing improved data retrievals in seasonal sea ice covered seas. We have shown the importance of directly considering the correct satellite mode of operation in the Arctic Ocean where SAR is the dominant operating mode. Our new data are of specific use for wave model validation close to the sea ice edge and is available at the link in the data availability statement.

**Keywords:** CryoSat2; waves; oceanography

## 1. Introduction

The climate of the Arctic Ocean is influenced by the presence of sea-ice—the frozen ocean surface that sits between and dominates atmosphere and ocean interactions. The past 20 years have seen reductions in the summer sea ice extent of approximately 10% per decade [1,2], further revealing the open ocean surface of the Beaufort and Siberian Seas. Ocean surface gravity waves generated by the wind are a key feature of the atmosphere-ocean interface over the global ocean controlling atmosphere-ocean exchanges of momentum and heat [3]. Ocean gravity waves interact with the sea ice that floats upon the ocean surface throughout the Marginal Ice Zone (MIZ): a region potentially 100 s of km wide [4] typified by increasing ice floe diameters and decreasing wave heights further into the sea ice pack and defined by sea ice concentrations of between 15% and 80%. The interactions are complex due to the multiple physical processes at play: sea ice floe mechanical breakup [5], gravity wave attenuation due to the damping caused by movement of sea ice floes [4,6], the spectral non-linearity of wave attenuation [7] and the differing wave attenuation of newly formed grease and pancake ice compared to consolidated pack ice [8].

Models exist that endeavour to replicate these physical processes [9] giving the spectral wave climate across the polar oceans and through the sea ice pack. For the inclusion of

sea ice floes within a wave model, the presence of sea ice stores and dissipates mechanical energy, typically represented as a viscoeleastic medium [10]. Ref. [11] asses the performance of a variety of wave model parameterisations against observations in the north west Atlantic in the presence of sea ice and ocean currents showing the importance of wave scattering by sea ice floes in the MIZ. The validation of such models is challenging due to the limited availability of wave data within pack ice, and the rapidly changing wind and thus wind driven wave state. Ref. [12] contrast differing satellite sea ice concentration data, showing how the difference between them alters hindcasts by the WaveWatch 3 wave model for 2018 in the Chukchi Sea. They show that despite model uncertainty for the complex physics in wave ice interaction, the concentration and extent of sea is a leading order influence in the modelling of wave height propagation through the sea ice pack.

Whilst there has been a global increase in ocean wave heights [13], the trend in the Arctic Ocean is inconclusive [9]. The larger expanse of summer Arctic open ocean allows for increased available wave fetch though wind swells (wind driven ocean surface waves) when encountering the sea ice edge cause the break up of sea-ice floes and the damping of the gravity wave spectrum. Forecasts and measurements of surface waves are critically important for operations at sea and understanding the interaction between surface waves and coastlines is of pressing concern: global sea level rise has lead to increased coastal erosion [14], and positive trends in global wind speed have driven trends in the most extreme wave heights [15].

Earth observation satellites have been in orbit for nearly five decades [16] carrying downward facing radar altimeters to study the height of the ocean surface. The shape of the radar impulse response can be analysed to asses the roughness of the ocean surface and thus the height of ocean swell waves. Successive satellites have carried altimeters and their data have been incorporated into continuous coherent global data sets on ocean dynamics and waves [17,18]. Additionally over this period there have been continuous advances in the accuracy of the radar instruments, the satellite tracking and the theoretical interpretation of the raw retrieved data. However the form of the radar instruments has not altered greatly: a normal incidence, pulse-limited altimeter with a circular 1 m diameter antenna. CryoSat-2 (CS2) was launched on 8 April 2010 [19] into a highly inclined 92° retrograde orbit giving coverage up to 88°N/S. It carries the bespoke SAR (Synthetic Aperture Radar) Interferometric Radar Altimeter (SIRAL) instrument which has three operating modes: as well as operating as a conventional pulse limited radar in its Low Resolution Mode (LRM), it can employ along track SAR processing to increase along track resolution, and in full SAR Interferometric (SARIn) mode to additionally record the across track slope. While LRM mode continues the data coverage of past and future ocean surface altimetry, SAR mode gives increased information on sea-ice covered seas and SARIn model is used to investigate the elevation and movement of ice sheets and glaciers.

The wave state of the ocean surface is characterised by the Significant Wave Height (SWH), described as the characteristic height of the random waves in a sea state as observed from a ship and defined as $4\sigma_s$ where $\sigma_s$ is the standard deviation of the sea surface elevation [20]. The use of existing LRM equivalent algorithms to retrieve SWH from CS2 has been performed by Radar Altimetry Database System (RADS) [21] for both LRM and pseudo-LRM (pLRM) modes (a conversion from the recorded SAR retrievals to theoretically coincident pulse limited echo [22]) and incorporated into large consistent wave data sets and models [9,17]. However CryoSat-2 operates primarily in SAR mode for the Polar seas, a region that has had significant change to the open ocean state over the period of CS2 operation [9], with reduced summer sea ice extent. The theory to retrieve SWH from SAR mode exists, and we present here an application of this theory to retrieve ocean surface information over the full CS2 period in order to improve the quality and coverage of observational SWH data in polar seas. The data processing chain presented in this paper can also be applied to other satellite missions and to cover the global oceans. The instrument specifications described in Section 2.3 can be adapted for the SAR instrument on Sentinel 3

for example, and the global LRM mode data from CS2 can be processed to give SWH data in mid-latitudes.

A description of the CS2 data format accessed for processing and the final data end products are described in Section 2.1. Additional supporting data from analysis are described in Section 2.2. A description of the new semi-analytical retracker of [23] is in Section 2.3 and its adaptation for use in a CS2 LRM and SAR mode data processing chain is in Section 2.4. We present an in-depth look at the retracking process in Section 3.1. Comparisons to alternative CS2 SWH data are presented in Section 3.4, validation against in situ wave buoy observations is in Section 3.5. The description of the final validated Arctic processed data is given in Section 3.7, with additional Antarctic data in Section 3.8. Final conclusions are in Section 4. The new data are available at http://www.cpom.ucl.ac.uk/ocean_wave_height/(accessed on 20 August 2021).

## 2. Data and Methods

### 2.1. CryoSat-2 Altimetry Data

CS2 level 1-B baseline-D data from ESA are used (https://earth.esa.int/eogateway/missions/cryosat, accessed on 20 August 2021). Individual files are available for each orbital path and instrument mode. All data over 60°N are selected and identified as open ocean for pulse peakiness <4.0 (LRM), <6.0 (SAR) and for the multi-look echo power standard deviation <0.07 (LRM), <0.10 (SAR). These waveforms are downsampled from 20 Hz to 1 Hz taking the mean echo power after alignment using a leading edge threshold of 50% and 70%, for LRM and SAR modes, respectively, if fewer than 90% of the original 20Hz waveforms are classified as representing an echo from sea ice covered seas. The downsampling reduces the number of retracking operations and reduces the speckle noise in each range bin, although potentially smearing the leading edge of the echo and producing artificially higher SWHs. However we find that our 1Hz processed record performs well in relation to coincident 20Hz records from our own processing (Section 3.1) and from the GPOD system (Section 3.4). Each CS2 level 1-B data file that contains any open ocean identified SAR or LRM echo from the desired region is processed and saved in a matching SWH file along with the retracking fit.

### 2.2. Addtional Accompanying Data

To compare our new semi-analytical retracker we have sourced data from previous efforts at retrieving SWH from CS2. RADS data were obtained from http://rads.tudelft.nl/rads/rads.shtml (accessed on 20 August 2021) for the entirety of 2014, with co-located WaveWatch 3 data supplied alongside [24]. We have gathered SWH from the SAMOSA+ (SAR Altimetry MOde Studies and Applications) retracker though the ESA GPOD system (Grid Processing On Demand). Due to the computational time taken to process the SAMOSA+ data using the ESA GPOD system a shorter period of 2018-11 to 2019-4 was used. It was not feasible to process the complete CS2 time period using the GPOD system.

For the locating of open ocean surface within the seasonally ice covered Arctic we used NSIDC ice concentration data [25]. We use daily ice concentration maps. For analysis in this study all areas where sea ice concentration is greater than 15% were omitted, although all ocean locations were processed.

To contrast the different satellite data sources we have endeavoured to find in situ wave buoy measurements of SWH above 60°N over the period of data collection. These buoys, documented in Table 1, are the NOAA NDBC buoys numbered 48,213 and 48,214, the Swift wave buoy array of [26] and the UK Met Office buoy K7 (https://www.metoffice.gov.uk/weather/specialist-forecasts/coast-and-sea/observations/164046, accessed on 20 August 2021).

**Table 1.** Wave buoy data used in this study, plus correlation to our new SWH. All buoy locations are under the SAR mode operation of CS2. Pearson correlation coefficients between our new SWH and the wave buoy measurements are listed. See Section 3.5 for details.

| Buoy Name | Period | Location | Correlation |
|---|---|---|---|
| Swift array | 28 July 2014 to 28 September 2014 | −144.5°E, 70.9°N<br>−159.9°E, 74.6°N | 0.77 |
| Met Office K7 | 2014<br>2015<br>2016<br>2017<br>2018<br>2019 | −4.5°E, 60.7°N | 0.89<br>0.91<br>0.97<br>0.96<br>0.96<br>0.96 |
| 48,213 | 1 September 2012 to 7 October 2012<br>1 August 2013 to 31 October 2013 | 164.1°E, 71.5°N | −0.43<br>0.14 |
| 48,214 | 1 September 2014 to 11 October 2014<br>1 August 2014 to 8 October 2014<br>14 July 2015 to 12 October 2015 | 165.2°E, 70.9°N | −0.35<br>0.51<br>0.5 |

### 2.3. Semi-Analytical Model

To retrieve information on the SWH of the open ocean surface, we match the recorded radar reflection to an idealised semi-analytical waveform. The shape of the idealised waveform is a function of the reflected surface roughness, that for the open ocean surface is dependent on the height of ocean surface gravity waves. The waveform model is defined over the range of expected SWHs (0 to 10 m) and for the satellite's orientational parameters. Wingham et al. [23] provide a model of a synthetic aperture, interferometric satellite radar altimeter echo power and echo cross product. This model depends on many instrument specific parameters (including the instrument transmit pulse $p_t(t)$) and the knowledge of the instrument location (range, inclination and velocity), which when combined with assumptions about the Gaussian nature of surface scattering $s(z)$ gives a large semi-analytical convolutional form of the echo power cross product.

For the retrieval of surface gravity waves the ocean surface is considered as a flat surface from which theoretical radar responses can be derived [27]. Individual recorded radar echo power waveforms $P(t)$, where $t$ is the time scale of the recorded echo power, can be described mathematically as the convolution

$$P(t) = p_t(t) * P_{FS}(t) * s(ct/2), \tag{1}$$

where $p_t(t)$ is the transmit pulse, $P_{FS}(t)$ is the flat surface impulse response and $s(ct/2)$ is the distribution of surface scatterers for $c$ the speed of light in a vacuum. The radar pulse transmitted from the SIRAL instrument has a shape that can be approximated as a Gaussian function

$$p_t(t) = \frac{P_0}{\sqrt{\pi}} \exp\left[-\left(\frac{t}{\Delta t_p/2}\right)^2\right] \tag{2}$$

where $P_0$ is the transmitted pulse peak power (later solved for within Equation (6)) and $\Delta t_p$ is the duration of the transmitted pulse (see Table 2 for values used). The probability distribution of the ocean surface with uniform waves is given as the Gaussian:

$$s(z) = \frac{1}{\sqrt{2\pi}\sigma_s} \exp\left[-\frac{z^2}{2\sigma_s^2}\right] \tag{3}$$

where is $z = ct/2$ is the instrument altitude and $\sigma_s$ is the surface roughness standard deviation. As with measurement convention the significant wave height is taken as $4\sigma_s$.

The flat surface response for a single beam $P_{FS}^k$ (numbered $k$) of the synthetic aperture stack is given by [28] as a modified form of that of a pulse limited altimeter with

$$
P_{FS}^k(t, B) \approx \frac{\lambda^2 G_0^2 D_0 c \sigma_0}{32 \pi^2 h^3 \eta} H\left[t + \frac{\eta h \xi_b^{k2}}{c}\right] e^{-ik_0 B(\chi + \beta/\eta)}
$$
$$
\cdot \int_0^{2\pi} d\phi d[\rho_k \cos \phi - \xi_b^k] e^{ik_0 B \rho_k \sin \phi}
$$
$$
\cdot \exp\left[-2\left(\frac{(\rho_k \cos \phi - \mu)^2}{\gamma_1^2}\right.\right.
$$
$$
\left.\left. + \frac{(\rho_k \sin \phi - \mu - \beta/\eta)^2}{\gamma_2^2}\right)\right]
$$

(4)

where $\lambda$ is the carrier wavelength, $G_0$ is the antenna boresight gain, $D_0$ is the one-way gain of the synthetic beam, $\sigma_0$ is the backscatter coefficient, $\mu$, $\chi$ and $h$ are the satellite pitch, roll and altitude, $\beta$ is the across-track surface vector gradient, $\eta$ is the factor to account for the curvature of the Earth and $H[t]$ is the Heaviside step function. Furthermore to allow for the slant range correction

$$
\rho_k = \sqrt{\frac{ct}{\eta h} + \xi_b^k}
$$

where $\gamma_1, \gamma_2$ are coefficients to account for the ellipticity of the SIRALs antennas with

$$
\gamma_1 = \sqrt{\frac{2}{2/\bar{\gamma}^2 + 2/\hat{\gamma}^2}}, \gamma_2 = \sqrt{\frac{2}{2/\bar{\gamma}^2 - 2/\hat{\gamma}^2}}.
$$

The function $d[\rho_k \cos \phi - \xi_b^k]$ in Equation (4) as described by Wingham et al. is used to describe the beam gain for the SAR rectangular footprint window and the LRM footprint.

**Table 2.** Parameter values for the semi-analytical retracker.

|  | Description | Value |
|---|---|---|
| $\Delta t_p$ | Pulse duration | 3.52 ms |
| $G_0$ | Antenna gain | 42 dB |
| $D_0$ | SAR synthetic beam gain | 36.12 dB |
| $\xi_b^k$ | beam look angles | 51 |
| $c$ | Speed of light | $3 \times 10^8$ |
| $\eta$ |  | $1 + h/R_e$ |
| $R_e$ | Earth radius | $6.38 \times 10^6$ m |
| $\lambda$ | Carrier wavelength | $c/(13.58 \times 10^9)$ |
| $k_0$ | Carrier wave number | $2\pi/\lambda$ |
| $\bar{\gamma}$ | Antenna shape parameter | 0.0122 |
| $\hat{\gamma}$ | Antenna shape parameter | 0.0382 |

Taking the semi-analytical form of a multi-look idealised SAR or LRM power waveform from Equation (4), the echoes depend on up to nine different instrument and surface variables: $\mu, \chi, h, \xi, \eta, \nu_s, t_0, \sigma_0, \sigma_s$. In SAR mode, it is possible to reduce this to two parameters if one is only interested in SWH. Firstly, the antenna baseline pitch and roll can be taken from the CS2 product, as they are measured independently by three star trackers mounted on the interferometer bench, as well as the satellite altitude. Secondly, Wingham et al. [23] find that for all but the largest possible altitude rates, the effect of the radial component of velocity on the beam formation is negligible (see [29] for further description). Finally, if the backscatter is not of interest, the absolute power is not important and the waveform can be normalised in such a way to optimise the fitting procedure.

Wingham et al. [23] provide the equational form to take the full convolution into a usable numerical form. The full echo power convolution from Equations (1)–(4) is fully integrated over the beam count $k$ and range ring angle $\phi$. This integration results in a fully analytical form for the echo power for specific $\sigma_s$ and $t$ (amongst other parameters) that can then be repeated over the expected range of $\sigma_s$ and over the echo range $t$ to provide look up tables. We used the simplifications for the specifics of the CS2 instrument (ellipticity of the antennas, relative smallness of the satellite pitch $\mu$, roll $\chi$ and altitude $h$, as discussed above) to give analytical forms of $\mu$, $\chi$ and $h$ while containing $\sigma_s$ and $t$ within lookup tables that provide the four components of the cross product $\mathbf{H^m}[\sigma_s, t]$ for $\mathbf{m} = $ SAR, LRM, such that

$$
\begin{aligned}
P(\sigma_s, t, \mu, \chi, h) = {} & \left( 1 - 2\left( \frac{\mu^{\,2}}{\gamma_1} + \frac{\chi^{\,2}}{\gamma_2} \right) \right) \mathbf{H^m}_1[\sigma_s, t] \\
& + 8\left( \left( \frac{\mu^2}{\gamma_1} \right)^4 \mathbf{H^m}_{11}[\sigma_s, t] + \left( \frac{\chi^2}{\gamma_2} \right)^4 \mathbf{H^m}_{12}[\sigma_s, t] \right) \\
& + \triangle h \mathbf{H^m}_2[\sigma_s, t].
\end{aligned}
\tag{5}
$$

We compare this technique to that used by RADS [21] which retracks the LRM and pLRM data from CS2 that relies on previous conversion of the SAR waveforms to a pLRM waveform. These data can be retracked using the same retracking algorithm of [30]. This algorithm, through a series of approximations gives the flat surface impulse response (4) as a combination of scaled exponentials, that give the final return power function (1) as a series of error functions that can be rapidly calculated for any input parameters. Whilst this algorithm is computationally efficient, there are numerous simplifications, particularly for the SAR footprint impulse response of CS2.

An alternate closed form of an idealised SAR response was developed as part of the SAR Altimetry Mode Studies and Applications over Ocean, Coastal Zones and Inland Water (SAMOSA) project. These equations use approximations such as an antenna point source and linearising the product of the radar beam gain over the cross section of the antenna [31]. This model of a SAR waveform is available as part of the ESA GPOD platform for all CS2 and Sentinel-3 mission SAR mode data.

An alternate semi-analytical form of the echo cross product is also available from [32]. This alternate semi-analytical form gives improved accuracy over a greater range of antenna mispointing angles and relies on fewer of the simplifications discussed above. However, implementing the semi-analytical methods of [32] to create the lookup tables required for the operational retrieval of SWH is beyond the scope of this study.

### 2.4. Operational Retrieval of Significant Wave Height

The lookup tables $\mathbf{H^m}[\sigma_s, t]$ have been generated for 51 beam SAR echo and a LRM echo using the parameters in Table 2 in alignment with [23,29]. The table files were read into a python function object, allowing for linear interpolation between the table entries. The object also includes all fitting functions (Equation (6)). In addition to the lookup tables a weighting $w_i$ for each mode is used, defined as the reciprocal of the estimated variance in the echo power for each range bin number.

We systematically opened the level 1-B files selecting all ocean records over $60°$N. The data filtering and downsampling was performed as described in Section 2.1. The resulting open ocean normalised 1 Hz echoes $\widehat{P}(t_i)$ were then accessed over bin number $[28, 88]$ for SAR mode and $[18, 78]$ for LRM mode. The echoes were then matched to the idealised echo in Equation (5) using coincident satellite orientation values $(\mu, \chi, h)$ also taken from the level 1-B file, using the optimisation relation

$$
S(A, \sigma_s, t_0) = \frac{\sum_i w_i \left( AP(\sigma_s, t_i - t_0, , \mu, \chi, h) - \widehat{P}(t_i) \right)^2}{\sum_i w_i}
\tag{6}
$$

where $A$ is the echo power scale factor (analogous to $\sigma_0$) and $t_0$ is the normalised wave range adjustment (equivalent to sea surface elevation). Minimising $S(A, \sigma_s, t_0)$ retrieves optimal values for $A, \sigma_s, t_0$ with all values and the resulting best fit value of $S$ recorded. We took all cases of $S < (2 \times 10^{-4}, 2 \times 10^{-3})$ for SAR and LRM modes to be cases of acceptable wave fit with these results included within the analysis in Section 3. These parameters were chosen to balance the data quality and coverage for both modes separately [29]. The filtering removes 25% of SAR mode data and 5% of LRM mode data. The lower error fit limit and increased number of removed data for SAR mode is due to no sea ice mask being used at this stage in the processing chain. The majority of discarded SAR mode data are from ice covered regions.

For the analysis of the Arctic wide SWH data fields were gridded onto a 100 km polar stereographic grid. The grid was chosen to allow for the maximum useful data coverage for a meaningful daily data field. Due to the orbit characteristics of CS2, greater data coverage is possible for a longer time period, though for SWH, daily variability in wind speeds restricts the limit on meaningful time resolution. We used the 15% contour from the NSIDC daily ice concentration plots to remove values retrieved from within the sea ice pack. All SWH records in each day (00:00 to 23:59) that pass the fit tests were collected into grid bins with the mean taken. Furthermore, recorded are the data record count for both modes in all grid cells.

## 3. Results

We present the details of our efforts to use the semi analytical retracker to extract SWH from the entirety of the CS2 data record 2011–2019. First of all we give an in-depth example of the wave fitting for a particular CS2 track in Section 3.1. We then contrast our results to existing satellite and model data, Section 3.4, and in situ buoy data, Section 3.5. The full Arctic data set is given in Section 3.7, with additional Antarctic data in Section 3.8.

### 3.1. Echo Fit Examples

Here we give details on the fitting of the idealised waveforms to the SAR and LRM echoes. When looking at the matched waves in detail, Figure 1, the semi-analytical echoes make a close match to the 1 Hz downsampled echos. Matching all three parameters $(A, \sigma_s, t_0)$ makes a closer fit than when matching $\sigma_s$ alone. Comparing the down-sampled pLRM to the coincident SAR in Figure 1a,b, there is a similar waveform fit, with a close match in retrieved SWH except for the red example. This is the case for the all the SWH retrieved from this track, see Figure 2, with a mean fit error of $5.7 \times 10^{-3}$ for pLRM 2.7 $\times 10^{-4}$ for SAR and $6.7 \times 10^{-4}$ for LRM.

For the example SAR echos (Figure 1) increasing from fitting $\sigma_s$ only, to fitting all of $(\sigma_s, A, t_0)$ greatly reduces the fit error from order $10^{-3}$ to $10^{-4}$ (Figure 3) greatly improving the reliability of the retrieved SWH. For comparison Figure 3 also contains the three parameter fit of every individual 20 Hz echo power waveform. The 1 Hz three parameter fit has a lower error than the 20 Hz waveforms throughout Figure 3 with the 1 Hz retrieved values for $(A, \sigma_s, t_0)$ at the centre of spread of 20 Hz retrieved values. In order to produce a full and usable SWH data set covering 2011–2019 the data retrievals from CryoSat2 had to be downsampled from the level 1-B 20 Hz record. Data at this frequency would have resulted in a dataset that was far too fine for use. We tested both 1 Hz and 20 Hz waveforms with Figure 3 showing how the 1 Hz waveform gets a more stable inversion. In particular we found that noise about the 20 Hz waveform peak caused the greatest variation in retrieved parameters. At present we are not confident that the inversion scheme is robust at 20 Hz and before further investigation resolves this, we therefore chose to restrict our analysis and inversions to 1 Hz averages.

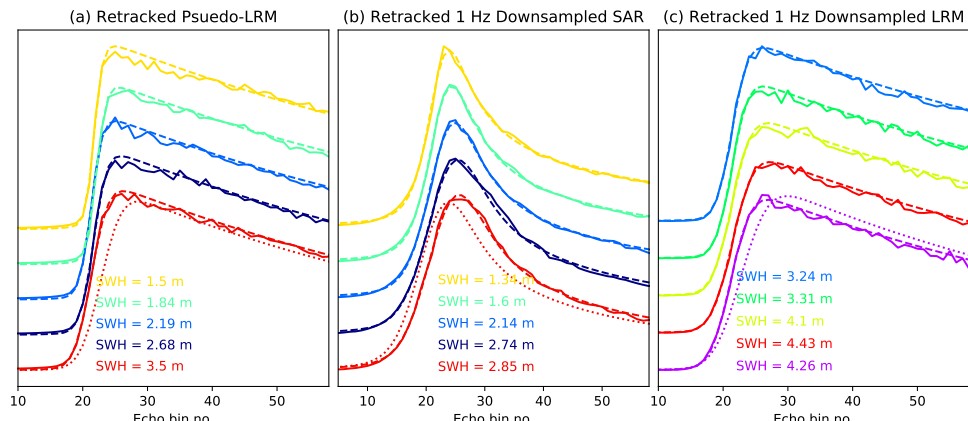

**Figure 1.** Example power waveform fit for (**a**) pseudo-LRM echoes (**b**) coincident 1 Hz down samples SAR echoes and (**c**) 1 Hz downsamples LRM echoes. Note that plots a and b originate from the same level 1 SAR retrievals. Each waveform (solid line) is plotted alongside an optimised 3 parameter fitted semi-analytic wave (dashed line) with the associated SWH listed . All waves are normalised by peak power and plotted at increasing height for clarity. The bottom most wave also has a 1 parameter (SWH only) matched wave plotted alongside (dotted line). The location of each wave is plotted in the same colors in Figure 2.

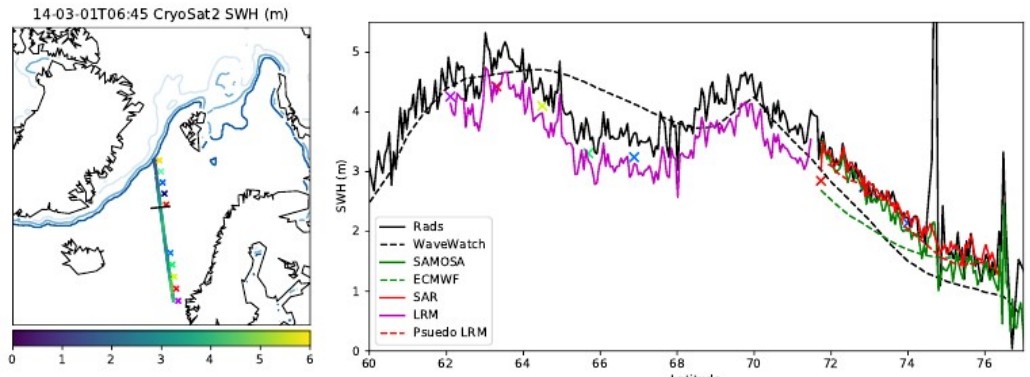

**Figure 2.** Significant wave height and ice concentration 6 September 2014 12:11. To the left is the map of the Arctic with NSIDC ice concentration contours of 15%, 50% and 85%. We plot our retrieved SWH from SAR and LRM modes, with the transition indicated by the black line. The coloured crosses correspond to the location of the echoes plotted in Figure 1. To the the right is the detailed SWH for the track from our new data (SAR and LRM modes), RADS (pseudo-LRM and LRM modes), SAMOSA (SAR mode) and the WaveWatch 3 and ECMWF wave models.

The fit error is low for the one and two parameter fit for waveforms numbered 2300 to 3200 with a similar SWH for all. We attribute this similarity to the similar wave power scale $A$ to the default value of $A = 1.0$ for these waves, while the retrieved leading edge $t_0$ is variable about the default value associated with wave bin number $t_0 = 28.0$.

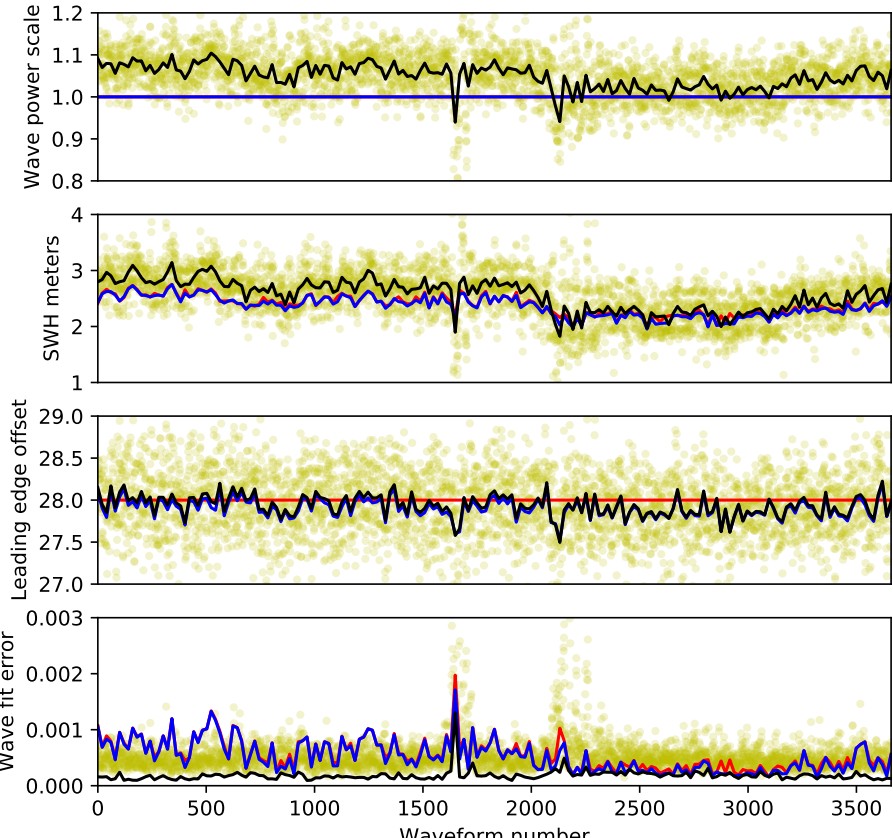

**Figure 3.** Comparison of the three methods of waveform fitting for 1 Hz downsampling over a section of satellite track in the North Atlantic. Red: fitting of $\sigma_s$ (SWH) only, blue: $\sigma_s$ and $t_0$ (waveform leading edge), black: $\sigma_s$, $t_0$ and $A$ (waveform power scale). These results are compared to a three parameter fit of each individual 20 Hz waveform (yellow scatter). Retrieved (or default values) of wave power scale, SWH and waveform leading edge are plotted, along with the fit error between the the original and optimal semi-analytical waveform.

### 3.2. North Atlantic 2014-03-01

Figure 2 shows an in-depth look at the retracking of radar echoes in the North Atlantic on 2014-03-01 shortly after 6am. We see a clear correlation between our retrieved LRM and SAR mode SWH with a linear decrease in SWH in both modes from 70°N to 76°N. Our retrievals from the pLRM waveforms show close correlation to those from the SAR mode. The comparison RADS data has the same spatial pattern of SWH with peaks of ≈5 m SWH at 61°N and 70°N. The WaveWatch 3 data (that was supplied alongside the RADS data) has a similar magnitude of SWH to the RADS data with less well correlated spatial pattern of peaks. For SAR and pLRM modes the RADS data are consistent to our SWH. For LRM mode the RADS SWH is approximately 0.34 m higher. While there are discrepancies between our new SAR, LRM, and pLRM, and RADS LRM retracked SWH in Figure 2, this single example is insufficient to draw conclusions from. In Section 3.6, we investigate the bias between our new SAR and LRM processing against examples from [17] and RADS data. There are likely to be further biases between these data and our retracked pLRM examples, though this is beyond the scope of this study as we focus on the construction of a full coherent dataset. The differences to pLRM do not just depend upon the assumptions and representations within the semi-analytical model used here as there is the issue of range walk compensation used within the creation of the pLRM waveforms from the base SAR echoes [22].

### 3.3. Central Arctic 2014-09-12

Figure 4 shows our CS2 retrieved SWH in comparison with ice concentration data. This track was chosen as it shows the changing sea ice concentration intersecting the open ocean wave state. The SAR mode shows a lower SWH east of Iceland which increases northward before decreasing before reaching the ice edge. A SWH of 1.7 m is found at a latitude of 84°N where the satellite path briefly intersects an area of open ocean. A SWH of 1.8 m is found in the East Siberian Sea beyond the sea ice edge. There is a consistent result from all three retracking methods for this example, except for the portion in LRM mode where the RADS SWH is approximately 0.16 m higher than ours. The SAMOSA data has strong visual correlation to ours for the SAR model. Both the WaveWatch 3 and ECMWF data correspond closely to the CS2 data for the North Atlantic with greater differences closer to the ice edge.

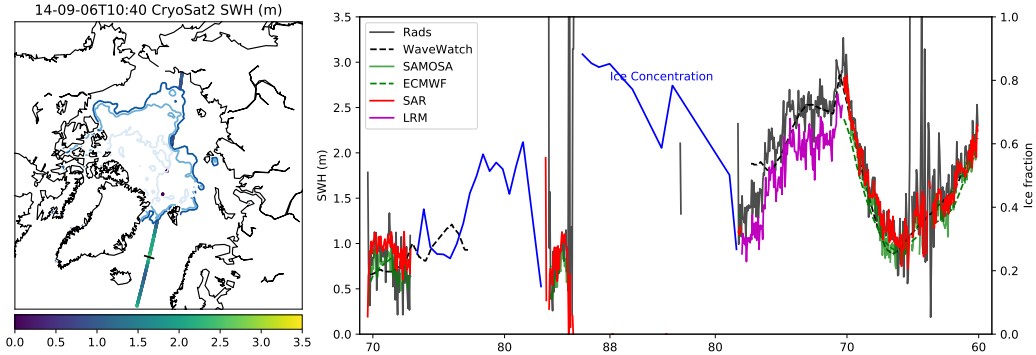

**Figure 4.** Significant wave height and ice concentration 6 September 2014 12:11. To the left is the map of the Arctic with NSIDC ice concentration contours of 15%, 50% and 85%. We plot our retrieved SWH from SAR and LRM modes, with the transition indicated by the black line. To the the right is the detailed SWH for the track from our new data (SAR and LRM modes), RADS (pseudo-LRM and LRM modes), SAMOSA (SAR mode) and the WaveWatch 3 and ECMWF wave models. For the regions where the orbit track intersects the sea ice cover the ice concentration is plotted. Note that the plotted latitude is symmetrical about 88°N, the northern limit of the CS2 orbit.

### 3.4. Alternate CS2 SWH Data Comparison

When comparing our new SWH product with existing CS2 SWH data and the accompanying modelled SWH there is a strong, near linear trend for all existing data (see top row of Figure 5). When looking at all data from 2014, for SWHs >4 m there is close correlation between our new SWH and the RADS SWH. On closer inspection we can see that our SWH collected from LRM mode is greater than the RADS data, while for SAR mode away from the sea ice edge there are small differences, with clear shapes in the NBIAS plots that matches the SAR/LRM collection areas (viewable in the SAMOSA plots which are for SAR mode only). For lower wave heights particularly in regions closer to the ice edge, there is high variability in the RADS data, with biasing up to 3 m higher than our data. As there is higher bias and decreased correlation near the ice edge, contamination of the RADS data with sea ice is likely. The WaveWatch 3 model data that was supplied alongside the RADS data has decreased correlation over the domain, with a particular decrease near the ice edge. Near the sea ice edge our new SWH is typically 20–30% greater than the WaveWatch 3. The bias map fro WaveWatch 3 matches the shape of the CS2 modes. For the SAMOSA data there is a trend of 0.88 to our new data. SAMOSA SWHs are typically are 0.3 m or 5% higher for SWHs greater than 6 m. This bias can be seen reflected in the open ocean and marginal seas. However, there is a close correlation between the two data sets. The ECMWF model shows a reduced correlation similar to the WaveWatch 3 model, but with a reduced spatial pattern of bias.

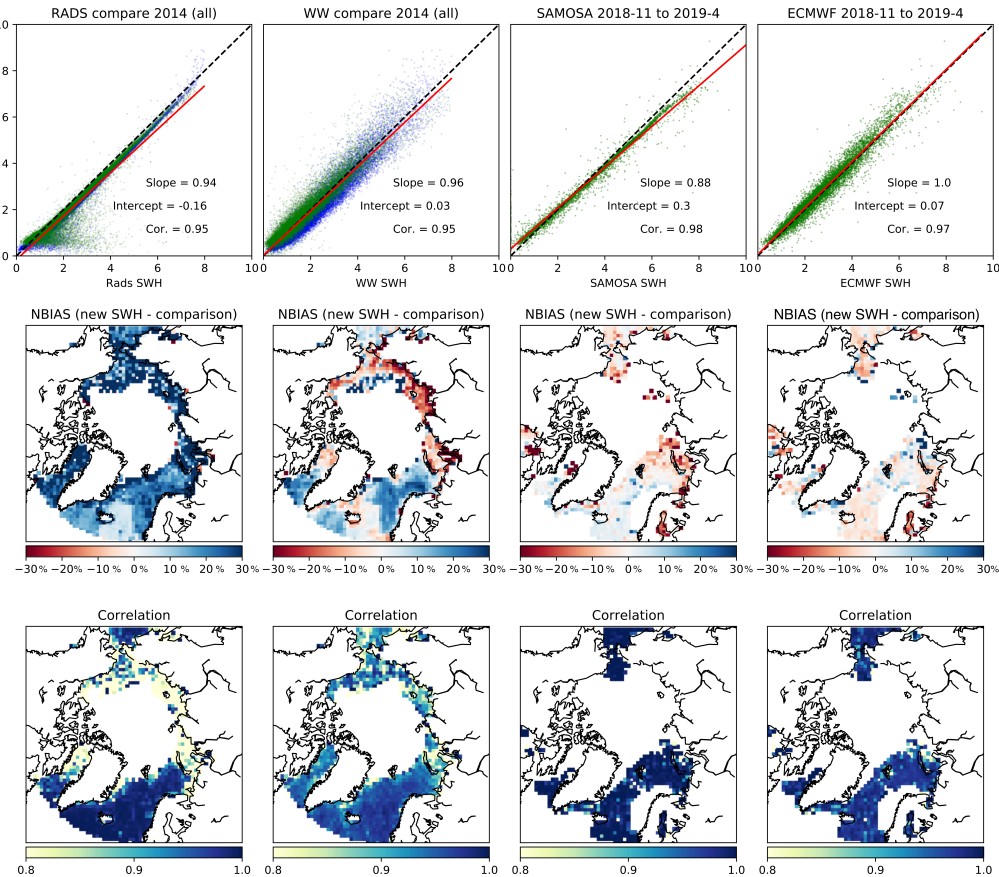

**Figure 5.** Comparisons between our new SWH data product and existing CS2 derived data sets and models. Comparisons are performed after binning the individual tack data onto a 100 km grid. The first column is for SWH from the RADS data. Column 2 is for the WaveWatch 3 model data. Column 3 is for the SAMOSA data. Column 4 is the ECMWF model data. The top row shows scatters comparing our data and the comparison data for both LRM (blue) and SAR (green) modes (SAR mode only processed by SAMOSA). The second row shows the Normalised Bias (NBIAS) (new SWH as a percentage of the comparison SWH) and the bottom row shows the correlation between our new SWH and the comparison data for daily averaged data for the time series listed above.

### 3.5. Buoy SWH Observational Validation

To validate our new SWH and aid in comparisons to models and alternate data presented in Section 3.4, we source all readily available wave buoy data within the satellite data domain. We find all satellite data records within 100 km of each buoy and take all the buoy wave height measurements within 3 hours of each passing satellite track. Due to the limited number of wave buoys in the Arctic we take a wider spatial area to [18] (100 km vs. 50 km). We found that reducing the spatial area for correlation to 50 km, the data correlation showed little improvement for a reduced number of coincident measurements. The pairs of measurements were used to calculate the Person correlation coefficient for each buoy record. The majority of buoy data accessed was from the 6 year time series of the Met Office Buoy K7 at $-4.5°$E, $60.7°$N. This buoy was within the SAR mode of operation for the whole time series. The top line of Figure 6 shows the comparison between the buoy record and our and RADS retrieved SWH from CS2 data coincident to the buoy location over the period 2014–2019. Our new SWH has an improved correlation compared to the RADS (0.88 to 0.86) and reduced bias, with our new SWH at +11% and RADS at +17%, Additionally SAMOSA data were also gathered for the whole 2018 for further comparison. The new SWH and RADS perform better for this year with correlations of 0.96 (0.97 for SAMOSA). The bias for 2018 is less than for the whole 2014–2019 time period with our new SWH at +7%, RADS at +12% and additionally SAMOSA data at +7%.

For the central Arctic two data sources were available. First is from the NOAA NDBC archive with records from fixed oil and gas platforms. These data are available for a few summers, with 2014 presented here in row three of Figure 6. This data source has times of good correlation (2014-08-3 to 2014-08-24), but also times where there is little match (2014-09-07 to 2014-09-21). Due to this both our new SWH and RADS have a correlation of around 0.5. The bottom row shows the buoy data collected as part of the MIZ project in the summer of 2014. These buoys are floating with regular contact with the ice edge (for example between 2014 and 09-07 and 2014-09-18), which presents a challenge for co-locating CS2 paths. The comparisons used here were made using our 100 km gridded SWH. The buoy presented here, shows good correlation with our new SWH (correlation of 0.77) while the RADS data at this location has no identifiable correlation.

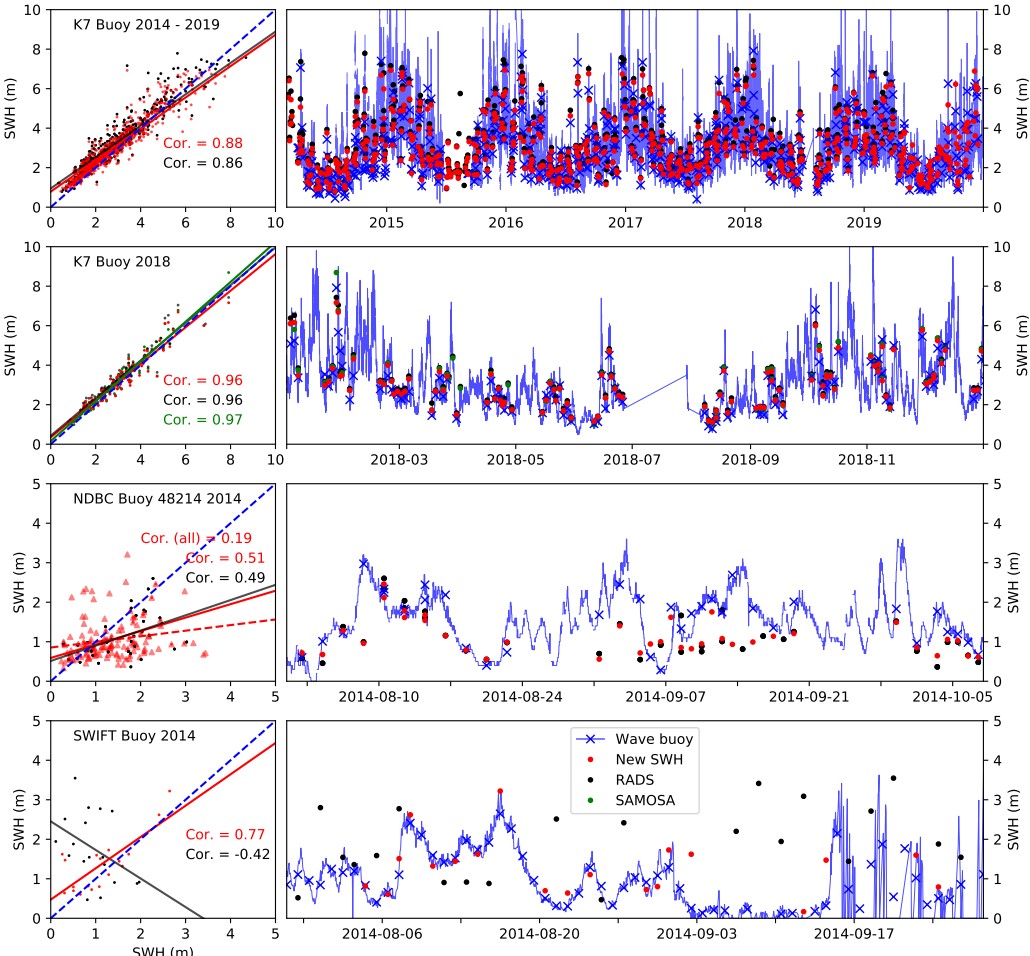

**Figure 6.** Comparison of our CS2 derived SWH and other satellite products. The top row is Met Office buoy K7 2014 to 2016 and our SWH and RADS data. The second row is a closer look at buoy K7 over 2018 with additional SAMOSA data. Row three is NDBC buoy 48,214 in 2014-(08 to 10). The bottom row is for the Swift floating wave-buoys deployed in the Beaufort sea in 2014-(08 to 09). The blue lines are the complete buoy wave height record, with crosses indicating the 6 hour mean value coinciding with a satellite record within 100 km of the buoy location.

### 3.6. Mode Bias Corrections

The retrieval of SWH from the two modes of satellite operation considered can be compared by considering locations where the satellite changes mode over the open ocean. In these locations when gridding the SWH onto the used 100 km resolution grid the SWH from both modes are likely to be considered for a grid cell. By considering the mean LRM derived SWH to the mean SAR SWH from such grid cells a mode to mode bias can be

calculated. Approximately 10,000 such locations were found for the daily collected data over the 9 year of processed SWH. Measurements are ignored if there are fewer than 4 LRM or SAR retrievals. Figure 7 shows correlation between mode offset and average retrieved SWH and the seasonal cycle of mode offset. The left panel shows a clear correlation between the bias and the wave state. For SWH <5 m, the SAR mode gives a lower SWH, with a higher value for SWH >5 m, although there is wider distribution for higher SWH. There is also a seasonal development with a lower offset in summer when wave heights are typically lower. [17] incorporate CS2 data into their combined data set using the RADS data product. As [17] have performed extensive validation of the RADS SWH data product to a long time series of in situ observations, comparing our new SWH data product with the corrected RADS data enables to evaluate the performance of our new SWH against the current scientific standard. The RADS SWH data, $s$ is adjusted to a corrected $SWH_C$ using a 3rd order polynomial with

$$\text{SWH}_c = \begin{cases} (0.4889 + 0.4712s + 0.1546s^2 - 0.0145s^3) & s \leq 2.45 \\ (-0.1057 + 1.0058s) & s > 2.45 \end{cases} \quad (7)$$

in order to correlate various satellite measurements with a buoy measurement archive. The correction given by Equation (7) is displayed as the green line in Figure 8. Figure 6 shows that our SAR mode SWH is highly correlated with in situ wave buoy measurements, while 7 shows that our new SWH has an emergent bias between LRM and SAR modes which can be used to adjust the SWH $SWH_L$ retrieved in LRM mode to the corrected $SWH_{Lc} = 0.38 + SWH_L/1.07$ by using the slope and intercept in Figure 7. We then compare our adjusted SWH (noting that the new SWH collected in SAR mode remains unchanged) to the corrected RADS data presented by [17] (second row of Figure 8).

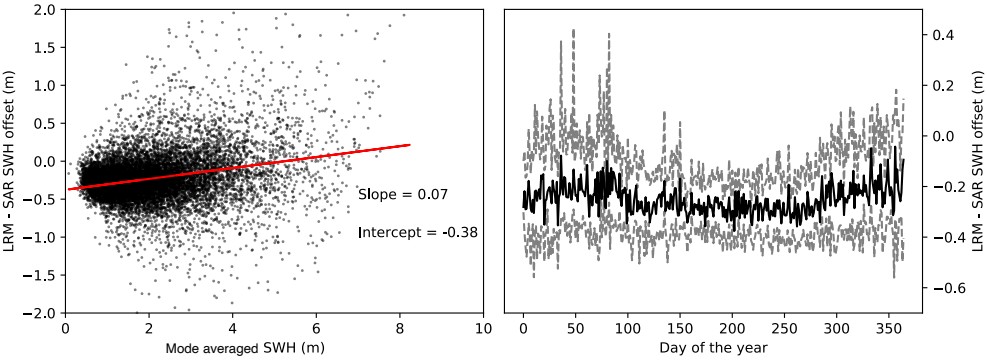

**Figure 7.** Difference between our retrieved SAR and LRM mode SWH 2011–2019. Plot to the left compares all coincident LRM and SAR mode SWH retrievals vs. the unscaled SWH. The plot to the right is the time series of 25th, 50th and 75th percentile of the LRM to SAR mode SWH offset.

The best fit slope becomes closer to unity for all cases shown in Figure 8. For LRM mode, in the scatter plot between the raw data (top left), there is a visible correlation between the scatter and the green correction line of [17] for SWH <1.5 m. The corrected version has no such shape and also has a best fit slope of near unity (0.97), and intercept of 0.01 m. As the analysis of [17] considers buoy data at lower latitudes, where CS2 operates in LRM mode, this result shows that our adjusted LRM SWH is in close correlation with existing corrected satellite SWH data. As our corrected LRM is matched to our raw SAR data this result (along with the improved slope to the SAR in the central column of Figure 8) shows that our SAR mode retrieved SWH is marked improvement on the pLRM processed RADS data, particularly in regions close to the sea ice edge.

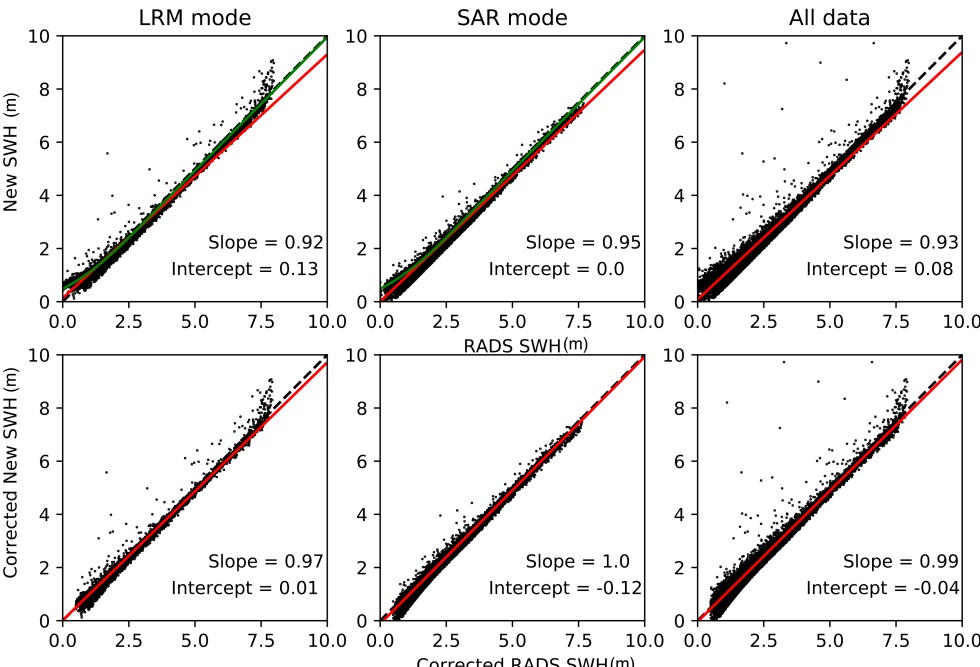

**Figure 8.** Detailed comparisons between our new SWH and the RADS product for data in 2014. Data has been collected onto a 100 km grid before comparison. Locations where RADS data are 0.5 m greater than our new data are omitted. The top row is for the raw data product, equivalent to that shown in Figure 5, the second row shows comparisons between the corrected RADS data and our new SWH with the mode bias removed. The columns separate between data collected in LRM and SAR mode. The green line in the top left panel represents the correction applied to the RADS data by [17]. The red lines are the linear best fit between our new data and the RADS data.

### 3.7. Full Arctic Data Set

Figures 9 and 10 summarise the full 9 year time series of CS2 SWH. Figure 9 shows that the higher wave heights are found in the North Atlantic during the winter months. This region shows a seasonal cycle with mean 4 m SWH during winter months reducing to 1–1.5 m during the summer (June to August as shown in the shaded region in Figure 10). The other perennial ice free regions—Baffin Bay, Barents, Kara, and Bering Seas—show a similar seasonal cycle with lower average SWH. The regions with a seasonal ice cover—Laptev, Beaufort and East Siberian Seas—are ice covered during the winter with no waves. From March to April onwards there is a slow increase in wave height as the sea ice retreats until there is a maximum from September to October before the sea ice advances once more. The Central Arctic and Kara Sea show a mixture of both cycles as they contain both perennial open ocean and seasonal ice.

The North Atlantic sector shows the winter of 2015 having the highest average SWH, matched by the a peak in the Barents Sea. However the Kara Sea had lower wave heights this season, and both it and Baffin bay had higher waves during 2017, not matched by the North Atlantic. All regions with perennial open ocean have higher waves during the winter. All regions have similar average wave heights near 1 m during the summer.

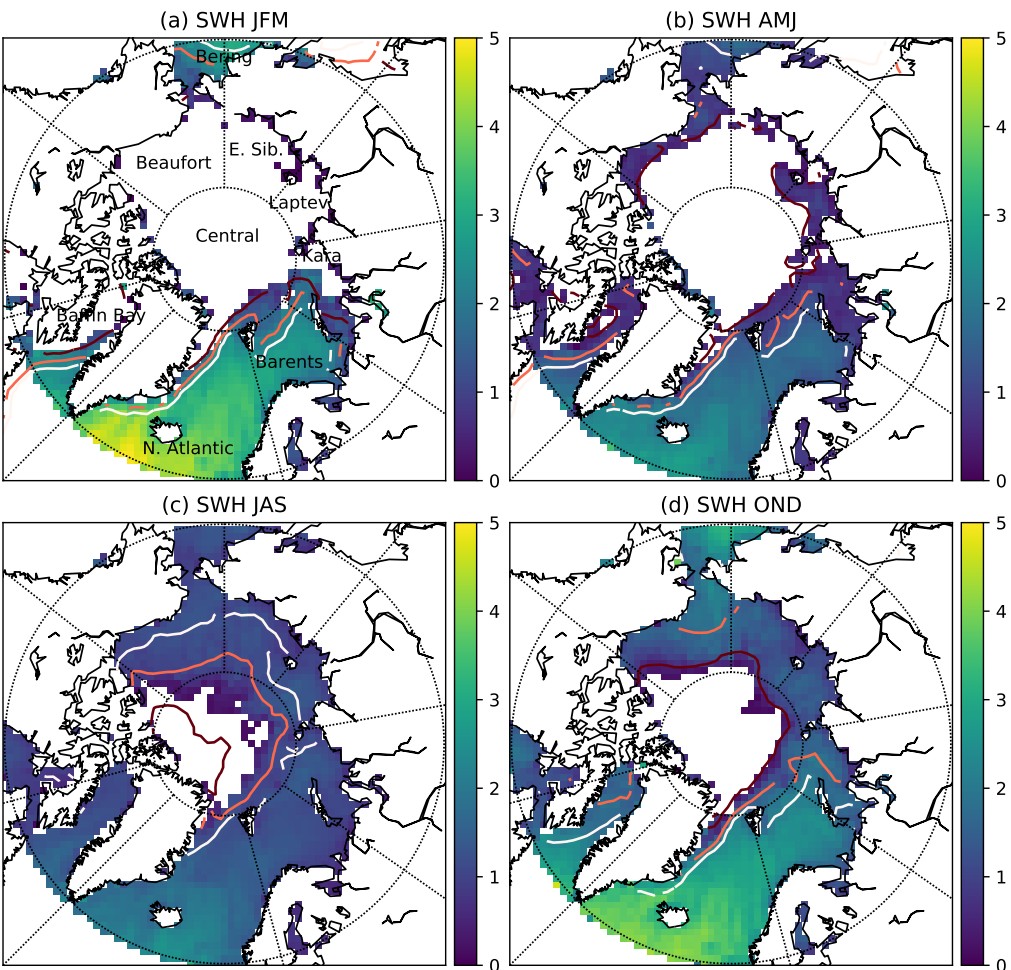

**Figure 9.** Climatological maps of retrieved SWH (m) for (**a**) Jan. Feb. Mar., (**b**) Apr. May. Jun., (**c**) Jul. Aug. Sep., (**d**) Oct. Nov. Dec. for the years 2011–2019. Plot (**a**) indicates the regions used in Figure 10. Overlaid on each plots are the average contours of 15% 50% and 85% sea NSIDC ice concentration for that quarter. Note that the Beaufort region also encompasses the Chukchi and Bering Seas.

Here we compare the extreme wave events apparent within the data set by looking at the height of the 95th and 99th percentiles for the winter period show in Figure 11. Whilst the 50th percentile shows a similar pattern to the mean seasonal maps (Figure 11) the 95th and 99th percentiles show the distribution of high wave events. To the south west of Iceland there are high SWH's apparent in the 95th (8 m) and 99th percentiles (at or even above the 10 m threshold of our retracker), indicating regular high swells. In contrast in the nordic seas see near 10 m waves in the 99th percentile but only 6 m for 95th, showing the high but rare large swells in this area.

In the central Arctic, the majority of ice free Beaufort sea remain sheltered, with all SWHs lower than 3 m and most less than 2. For the Chukchi and East Siberian seas, a number of rare events occur with swells from 4 to 6 m.

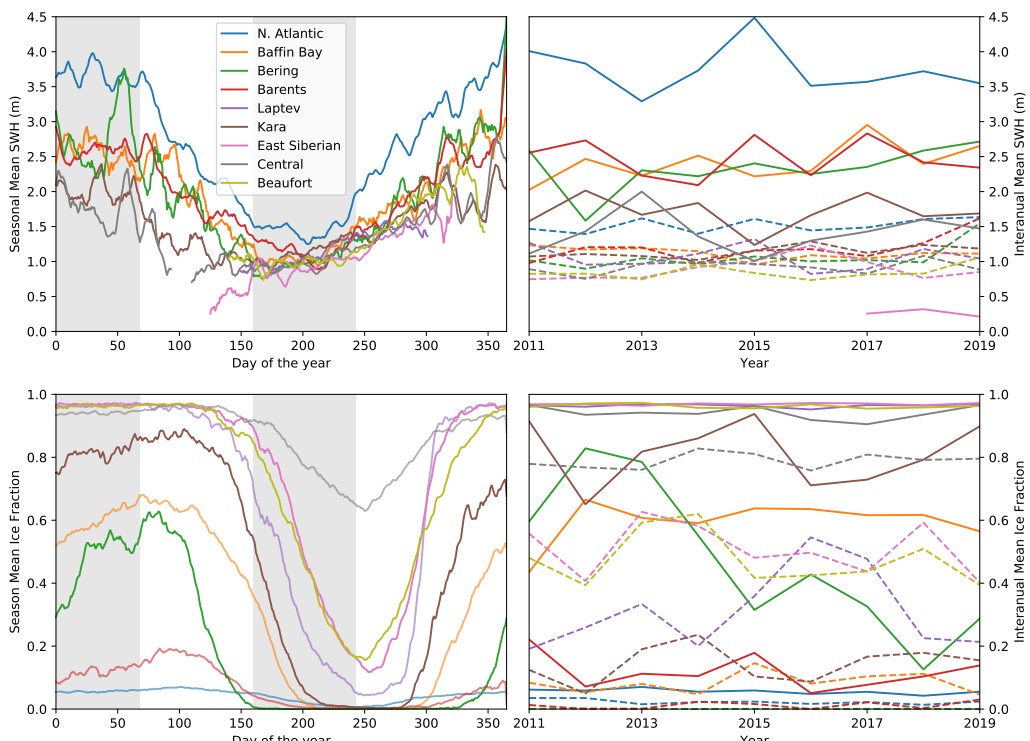

**Figure 10.** Seasonal cycle (**left**) and interannual variability (**right**) of SWH (m) and ice concentration for the regions indicated in Figure 9a. The interanual variability is for the days 01-01 to 03-10 (solid lines) and days 06-10 to 08-30 (dashed lines) periods shown by the grey bands on the seasonal cycle.

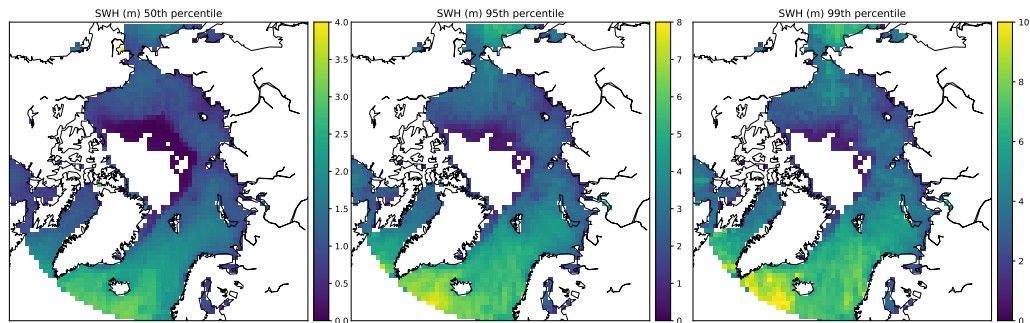

**Figure 11.** The 50th, 95th and 99th percentile of SWH for the entire period 2011–2019.

### 3.8. Full Antarctic Data Set

Here we present the 9 year time series of SWH for the southern ocean and marginal Antarctic seas. This data has been processed with same methodology as for the Arctic up to $-50°$S, but due to lack of Southern Ocean wave buoys, the detailed in situ comparisons of Figure 6 are not repeatable. The LRM mode data has been bias corrected in accordance to the methodology in Section 3.6 with an emergent LRM correction of $SWH_L c = 0.30 + s_L/1.07$ (see Supplemental Figure S1).

Figure 12 shows the Southern Ocean between $-50°$and $-60°$S has a strong season cycle in SWH peaking during July (see Supplemental Figure S2). The Southern Ocean and Bellingshausen and Admunsen sectors have a seasonal cycle similar to the North Atlantic sectors in the Artic, wave heights following the seasonal cycle in wind speeds. The Weddell Sea sector has the seasonaly ice covered sea characteristics, wave heights increasing with decreasing sea ice cover, with a peak in March. The Ross Sea, West Pacific and Indian Ocean sectors have cycles influenced by both sea ice cover and atmospheric conditions.

A notable characteristic for the data shown in Figure 12, is the emergence of wave height data within the limits of 15% ice concentration. These data have been masked by

the NSIDC daily ice concentrations, so show the retrieval of wave height information from within the large polynyas that form with Antarctic sea ice.

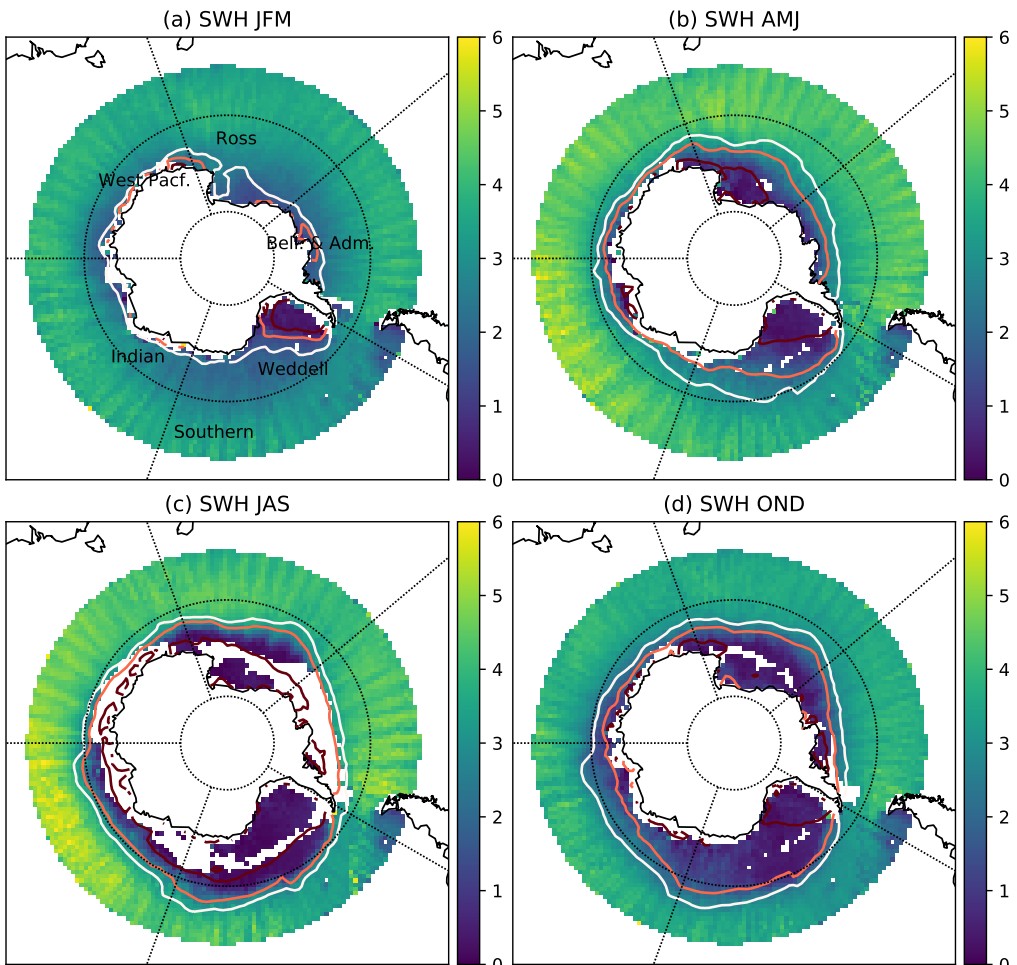

**Figure 12.** Antarctic climatological maps of retrieved SWH (m) for (**a**) Jan. Feb. Mar., (**b**) Apr. May. Jun., (**c**) Jul. Aug. Sep., (**d**) Oct. Nov. Dec. for the years 2011–2019. Plot (**a**) indicates the regions used in Figure S1 (Supplemental Material). Overlaid on each plots are the contours of 15% 50% and 85% sea NSIDC ice concentration for that quarter. Note that the Weddell, Indian, West Pacific, Ross and Bellingshausen & Admunsen sectors extend only to −60°, whilst the Southern Ocean region is over all longitudes..

## 4. Conclusions

We have processed the CS2 data archive from 2011 to 2019 to retrieve SWHs in the ice free polar ocean with a focus on the Arctic. These data are are available as 1Hz satellite records and combined into 100 km spatial and 1 day temporal resolution gridded product. We focussed on obtaining wave height information close to the sea ice edge in order to aid the development of wave models in the seasonally ice covered Arctic ocean.

The new semi-analytical retracker of [23] has accurately matched both pulse limited (LRM) and synthetic aperture (SAR) waveforms, processed with computational efficiency. The retracker is consistent with previous retracking attempts [21,31] and has given notable improvements in both coverage and data quality.

Compared to the RADS data set, there are marked differences in the two different modes of operation. Whilst the LRM mode uses the same base data, in SAR mode RADS uses a pre-processed pLRM. This difference along with our discovered LRM to SAR mode bias, give spatial inconsistencies for both data. There is a clear improvement with the

marginal seas, where the RADS processing chain does not adequately filter out sea ice contaminated open ocean radar echo waveforms and thus the accuracy of any retrieved SWH.

The developed processing chain allows for the accurate checking of our waveform fit to gather the cleanest and widest data coverage. Whilst our retrieved SWH show the closest correlation to the SAMOSA physical retracker obtainable through the ESA GPOD system, our dedicated SWH focussed processing chain allows for the processing of the entire CS2 time series, a task not possible using the GPOD system that only works for CS2 SAR mode. Note that the consistency of our 1 Hz downsampled waveform processing has the same data quality to the 20 Hz processed SAMOSA data.

Far from the influence of sea ice, in the North Atlantic, the comparison with buoy K7, Figure 6, shows consistency for all the retracking methods, with a slight improvement for our SAR specific new method and the SAMOSA data. In marginal seas our new data maintains its consistency to in situ observations where other products can become degraded or lose coverage. This data set was produced as part of the NERC Marginal Ice Zone project to address the lack of observational wave height data in marginal seas. As we have focussed upon the accurate analysis of CryoSat2 data in these locations and made sure to make our data are consistent with those produced for further ocean data, this data set makes a significant contribution to the validation of ocean wave models in polar regions. The data are available at http://www.cpom.ucl.ac.uk/ocean_wave_height/, (accessed on 20 August 2021) with a summary of the processing code at https://github.com/CPOMUCL/SWH, (accessed on 20 August 2021).

**Supplementary Materials:** The following are available online at www.mdpi.com/xxx/s1, supplemental Figures S1 and S2 are additional information of Section 3.8.

**Author Contributions:** Conceptualization, T.A., M.T. and H.H.; methodology, T.A. and H.H.; software, T.A. and H.H.; validation, H.H.; formal analysis, H.H., M.T. and J.L.; data curation, H.H. and A.R.; writing, H.H.; supervision, project administration and funding acquisition, M.T. All authors have read and agreed to the published version of the manuscript.

**Funding:** This research was funded by NERC grant number NE/R000654/1. MT acknowledges additional support from the NERC "PRE-MELT" (Grant NE/T000546/1) and "MOSAiC" (Grant NE/S002510/1) projects and from ESA's "CryoSat+ Antarctica Ocean" (ESA AO/1-9156/17/I-BG) and "EXPRO+ Snow" (ESA AO/1-10061/19/I-EF) projects. J.L. acknowledges support from the Centre for Integrated Remote Sensing and Forecasting for Arctic Operations (CIRFA) project through the Research Council of Norway (RCN) under Grant #237906.

**Data Availability Statement:** Our new data are of specific use for wave model validation close to the sea ice edge and is available at http://www.cpom.ucl.ac.uk/ocean_wave_height/, (accessed on 20 August 2021).

**Conflicts of Interest:** The authors declare no conflict of interest.

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
