# Peer review of "CryoSat-2 Significant Wave Height in Polar Oceans Derived Using a Semi-Analytical Model of Synthetic Aperture Radar 2011–2019"

_remotesensing, doi:10.3390/rs13204166_

Round 1
Reviewer 1 Report
CryoSat-2 significant wave height in the Arctic Ocean derived using a semi-analytical model of Synthetic Aperture Radar 2011-2019. This paper presents an interesting exploration of using a semi-analytical waveform model to retrieve sea surface wave height from CryoSat-2 data. This paper is well written and structured, the methodologies are well described, and result properly discussed. The paper can be accepted for publication subject to minor revisions.
Comments:
1. In this study, the waveform data from LRM and SAR were downsampled from 20 Hz to 1 Hz by taking the mean echo power. This process seems to help obtain more stable SWH values, as in Figure 2 the retrieved SWH from 20 Hz data is more scattered than that from 1 Hz. It’s suggested that more comparisons between 1 Hz and 20 Hz are needed as done in Figure 1.
2. Section 3.8 Full Antarctic data set: since the title has indicated that the study area is in the Arctic Ocean and the retrieved surface wave heights were validated in the Arctic Ocean not in the Antarctic Ocean, this section could be eliminated.
3. Line 196-198: Please check the grammar of the sentence.
4. Page 7 in Figure 1: “The location of each wave is plotted in the same colors in figure 31”, where is the figure 31?
5. Line 311-314, “We find all satellite data records within 100 km of each buoy and take all the buoy wave height measurements within 3 hours of each passing satellite track.” In this study, the SWH values are gridded at a resolution of 100 km resolution to obtain daily data. However, for validation, is it possible to use a finer resolution, such as 50 km or 25 km to reduce the spatial variation?
6. Page 12 in Figure 7: There are three lines in the right plot, please add more explanation to the three lines.
7. Page 13 Equation (7): The second row (−0.1057 + 1.0058)?
8. Line 358: How the equation SWHLc = 0.38 + sL/1.07 was obtained? Please clarify.
9. Line 446: “it’s” should be “its”?
Reviewer 2 Report
Please find my comments in the attached file.

Reviewer 3 Report
This paper provides an alternative method to estimate significant wave heights in an area where they are hard to obtain due to the large amount of contamination from sea ice in the radar echoes, which makes physical models fails. The method is validated using both other methods and in situ buoy data in the entire Arctic Ocean. The method is able to derive SWH estimates accurately and efficiently, even though the study area is challenging.
I enjoyed reading the paper. The method is well explained and the experiments are fitting and thorough. I just don’t understand the choice of SAMOSA2 vs SAMOSA+. You are choosing the fully analytical version of SAMOSA, which is bound to fail and provide fewer estimates in the Arctic, and you also mention it’s lower efficiency. I think it is an unfair comparison. You need to at least let the reader know that another, more appropriate, version exists, and tell them why you are moving forward with this version.
For consistency, you need to decide whether to write PseudoLRM, Pseudo-LRM, Pseudo LRM, PLRM, pLRM etc. Do the same with Level 1-b/B
And I think it would be suitable if you put the method in context with other missions and areas. Is it only relevant for CryoSat-2 and in the Arctic/Antarctic regions?
33: disapates -> dissipates
34: Missing space after .
41: , -> .
74: C2 -> CS2
75: PsuedoLRM -> Pseudo-LRM (PLRM)
77: Croysat-2 -> CryoSat-2
87: in depth -> in-depth
88: is -> are
93: atlimetry -> altimetry
94: update reference
98: wave -> waveforms
104: is -> in
106: section3.4 -> section 3.4
117: update reference
123: update K7 reference
126: Radar -> radar
130: et al -> et al.
133: instruments -> instrument
Table 1: New ->new
140: can described ->can be described
142: scatters -> scatterers
145: equations -> equation
Table 2: semi analytical -> semi-analytical
168+172: et al -> et al.
178: ellipsisity -> ellipticity
180: 4 -> four
182: Rads [20] who retrack -> RADS [20] which retracks
183+184: Psuedo -> Pseudo
184: use the same retracking -> be retracked using the
195: Sentinel -> Sentinel-3
202: 51 beam SAR echo -> 51 SAR echoes? Not clear.
212: 5 -> (5)
219: Why is more SAR mode data being removed?
Figure 1: psuedo –> pseudo
231: in depth -> in-depth
237: wave forms -> waveforms
242: Psuedo -> Pseudo
242: 1 -> Figure 1
242: wave form -> waveform
246: For the the -> For the , the just ->just the
250: that of -> than
252: waves -> waveforms
Figure 2: north -> North, comped ->compared?
257: in depth -> in-depth
259: in figure showing -> with
270: bias’ -> biases
277: Ice -> ice
282: he -> the
Figure 5: Why is the extent of ECMWF the same as SAMOSA?
339: gird -> grid
Figure 8: onto a 100 km before -> onto a 100 km grid before
372: summaries -> summarise
Figure 10: season -> seasonal, interanual ->interannual
Figure 12: What is Figure S1?
446: it’s -> its, in-situ-observations -> in-situ observations
